# Understanding the Impact of Social Networks on the Spread of Obesity

**DOI:** 10.3390/ijerph20156451

**Published:** 2023-07-26

**Authors:** Mark Tuson, Paul Harper, Daniel Gartner, Doris Behrens

**Affiliations:** 1School of Mathematics, Cardiff University, Cardiff CF24 4AG, UK; harper@cardiff.ac.uk (P.H.);; 2Aneurin Bevan Continuous Improvement, Aneurin Bevan University Health Board, Caerleon NP18 3XQ, UK; 3Employee Wellbeing Service, Aneurin Bevan University Health Board, Cwmbran NP44 8YN, UK; 4Department of Economy and Health, University of Continuing Education Krems, 3500 Krems an der Donau, Austria; doris.behrens@donau-uni.ac.at

**Keywords:** social networks, hybrid simulation, health care, behavioural OR, obesity, hyper-parameter optimisation

## Abstract

Previous research has highlighted the significant role social networks play in the spread of non-communicable chronic diseases. In our research, we seek to explore the impact of these networks in more detail and gain insight into the mechanisms that drive this. We use obesity as a case study. To achieve this, we develop a generalisable hybrid simulation and optimisation approach aimed at gaining qualitative and quantitative insights into the effect of social networks on the spread of obesity. Our simulation model has two components. Firstly, an agent-based component mimics the dynamic structure of the social network within which individuals are situated. Secondly, a system dynamics component replicates the relevant behaviours of those individuals. The parameters from the combined model are refined and optimised using longitudinal data from the United Kingdom. The simulation produces projections of Body Mass Index broken down by different age groups and gender over a 10-year period. These projections are used to explore a range of scenarios in a computational study designed to address our research aims. The study reveals that, for the youngest population sub-groups, the network acts to magnify the impact of external and social factors on changes in obesity, whereas, for older sub-groups, the network mitigates the impact of these factors. The magnitude of that impact is inversely correlated with age. Our approach can be used by public health decision makers as well as managers in adult weight management services to enhance initiatives and strategies intended to reduce obesity. Our approach is generalisable to understand the impact of social networks on similar non-communicable diseases.

## 1. Introduction

In 2020, the World Heath Organisation estimated that non-communicable chronic diseases (NCDs) accounted for 41 million potentially preventable deaths worldwide [1]. These included deaths from drug abuse, excessive consumption of food or alcohol, avoidance of exercise, and smoking. The explanations offered to account for individuals continuing to indulge in these behaviours include physical addiction, behavioural economics, environmental, cultural, and social factors, as well as inheritance and genetics. All of these factors are relevant in the case of obesity [2].

Similarly, the headlines from the World Health Organisation’s World Obesity Atlas [3] suggest that, in 2019, over 160 million years of healthy living were lost due to high BMI, more than 20% of all the lost years caused by NCDs. They predict that, by 2030, 1 in 5 women and 1 in 7 men across the planet will be living with obesity,

The Atlas also suggests that the greatest number of people with obesity now live in low- and middle-income countries, where they share a double burden of malnutrition and systems that are severely under-prepared and ill-equipped to effectively address obesity and its consequences.

The role played by social networks in the spread of obesity was highlighted by Christakis and Fowler [4] using longitudinal data from the Framingham Heart Study [5]. They demonstrated that obesity appeared to spread in a contagious fashion through social networks. The authors speculated that the spread might involve two mechanisms: firstly, homophily (the tendency of individuals to associate with others with whom they share similar traits), and, secondly, a change in individuals’ perceptions of what constitutes acceptable norms in terms of weight and obesity.

In this research, we explore the impact of social networks on the spread of obesity through a hybrid simulation (HS) in which the social network is replicated using an agent-based model (ABM) and individual health behaviour is modelled using a system dynamics (SD) approach. These are integrated in a single coherent framework and parameterised using longitudinal data from Health Survey England (HSE) in conjunction with a simultaneous perturbation stochastic approximation (SPSA) algorithm [6].

Our findings suggest that homophily and varying ‘norms’ are indeed factors in the spreading mechanism, as postulated in [4], and that, for the youngest population sub-groups, the network acts to magnify the impact of the external environment, whereas, for older sub-groups, it acts to reduce its impact. The magnitude of the network effect is inversely correlated with age. The simulated population proved surprisingly insensitive to significant changes in the external environment.

Our work contributes to the growing body of work in behavioural operational research, as reviewed and discussed in [7], building on and extending previous work in this area (e.g., [8,9]), and is novel in that it uses hybrid simulation to address a problem involving individual-health-related behaviour within the context of a social network. Other novel elements include simulating the social network using an emergent ABM based on social paradigms in order to create a stable but dynamic network. Optimisation/calibration of the HS is achieved with a stochastic approximation algorithm. We have also generated insights to inform the implementation of future obesity interventions.

The paper is organised as follows. Section 2 summarises our review of related work and provides the rationale for our approach. Section 3 describes the two key components of our model, and Section 4 details the hyper-parameter optimisation process used to verify, refine, and optimise the model’s parameters. This is followed by the key experimental results in Section 5. Section 6 provides discussions and conclusions.

An overview of the methodologies used to deliver our results is provided in Appendix A.

## 2. Related Work

### 2.1. Social Network Models in Health Care

Modelling social contact networks is often perceived as challenging. Bernstein and O’Brien [10] state the problem succinctly in their paper describing a stochastic ABM replicating a social network:


*“Researchers face a trilemma of inadequate data from real world datasets, statistical simulation models, and agent-based simulation models. Large-scale real world data sets are expensive to collect and difficult to obtain high fidelity ground truth for. Statistical models, such as Erdös–Renyi, Chung–Lu, and blockmodels, have parameters that are easy to specify and allow for simple replication of large-scale data sets. What is often missing, however, is the ability to encode narratives into the data because there is no sense of individual agents, just interactions between nodes.”*


In addition, social contact networks vary considerably in topology according to type. Newman and Park [11] compare a scientific collaboration network with that of a board of directors and obtain a variation of 58% in assortativity values. Anyone attempting to model a social network in the context of obesity must address a number of issues; apart from the studies already referenced, there is little or no information on the topology of ‘obesity’ networks; they may vary significantly in terms of that topology from other types of contact networks, and there may be significant variance amongst ‘obesity’ networks.

A review of 43 social network models used in modelling NCDs and the spread of disease identified three approaches: data-driven models, graphical models, and emergent models. The vast majority of these focused on contact networks rather than those enabled by electronic media.

Where real-world panel or longitudinal data exist, then using that data is perhaps the most credible option, but it does still raise a number of issues. The original research [4] used network data originally collected in relation to a study of heart disease. Each person was asked for the names of a maximum of four individuals who they would turn to for ‘advice’. These data were re-purposed to define a social network, creating at best a partial view of the actual network under consideration.

Graphical modelling approaches were the most popular (20). Three approaches were identified: Erdös–Renyi, Watts–Strogatz, and scale-free. They also have the ability to replicate the multiplexity (two individuals interacting with each other in different networks) inherent in social networks [12,13]. However they also share a number of issues when realising social networks over an extended timeframe:They require data to accurately represent the social network topography and multiplexity.The ‘realisation’ mechanism bears little relation to the social network constructs that create and maintain networks [10]. In these models, typically, a node’s tendency to form connections is based on probability and distance and does not take account of constructs such as homophily and propinquity.There is no obvious mechanism for exhibiting the dynamic behaviour that social networks exhibit over time, with individual connections being made and broken and sometimes remade. The only strategy identified was randomly ‘re-wiring’ connections.

These points are significant in the context of obesity because dynamic behaviour over extended timeframes (decades) is identified as a key element in the modelling process [14]. Additionally, our data were collected at the national level, and, thus, we have no specific data on network topography or the incidence of multiplexity. In the context of our research and the data we have available, graph models have less face validity and are perhaps best suited to applications where the time periods are relatively short.

Emergent models offer the most flexibility in simulating complex networks for disease modelling [15]. They are also able to deliver the required dynamic behaviour, usually using utility functions or stochastic rules. Reference [16] uses social theory to develop a model of disease spread in biological populations. Reference [17] takes this further in an exploration of social network stability. A set of homogeneous agents are given a set of behaviours and parameters that reflect the relevant social constructs, combining them to deliver a topographically stable network, which nonetheless exhibits the characteristic dynamic behaviour of ‘individuals’ within the network.

### 2.2. Hybrid Simulation in Health Care

Nianogo and Orah [18] identify the issues that may attend the use of hybrid models like ours, suggesting that the need for large data sets and significant computing power may prove problematic and that the models may be of potentially limited use elsewhere. Both [18,19] suggest that verification and validation are likely to be problematic.

### 2.3. Modelling Individual Behaviour

Achieving the research aims required an appropriate behavioural paradigm for the NCD, which could be operationalised in a simulation. A number of such paradigms were identified in the articles reviewed, and these are discussed below in the context of obesity.

The medical perspective on models of addiction addresses the behavioural patterns of obese individuals suffering from bulimia nervosa and binge eating disorder, although these represent a small percentage of obese individuals. It is less effective at describing the behaviours of the remainder of the population [20,21].

Behavioural economics (using utility or cost functions) and the theory of rational addiction are hindered by their implicit assumptions of rationality and logical process [22], often failing to consider the imperfect abilities and perceptions of the people to whom they are applied.

Navarro-Barrientos et al. [23] implement Ajzen’s Theory of Planned Behaviour [24] (TPB) in a dynamic (fluid flow) model to model simple weight loss scenarios. Aside from a range of health care applications [25,26,27], it has also been used for a wide range of other behavioural applications, from investigating academic dishonesty amongst business school students [28], through modelling consumer behaviour with respect to plastic waste [29], to investigating binge drinking behaviour amongst young adults [30].

Information as a dynamic parameter is another credible approach, with multiple examples in the literature and a strong body of current research primarily driven by investigation of diffusion and cascade processes in social media (e.g., [31,32,33]). These are effective where the main interest is the level of impact of the behaviour and require data to realise them. Where qualitative insights into the behaviour are also required or data are unavailable, then their value is reduced.

### 2.4. Conclusion and Contributions of This Paper

The models proposed in this paper can be categorised into and differentiated from the literature as follows. Simulation of social networks in health care modelling has previously used collected (actual) data, graph models, or geographical models. Of these, graph models are by far the most popular. In our research, we have chosen a novel emergent model to replicate and sustain the social network. The approach allows us to more closely replicate the social paradigms that drive and characterise social networks. This provides the opportunity for qualitative as well as quantitative analysis. Our approach also allows us to model a network that can be set or tuned to deliver a topography whose metrics match those of other observed social networks, is stable over an extended period, and exhibits the dynamic behaviour that is a critical element in the network effect when it comes to modelling the spread of obesity [4]. Our modelling is based on TPB, which is a well-established model for health behaviour applications. By extending the approach used by Navarro-Barrientos et al. [23], and implementing it as an SD model, we are able to embed it within a network model as part of an HS. This novel approach provides the potential for further qualitative insights [34].

Furthermore, our SPSA algorithm for optimising the simulation is in line with the recommendations of [35]. However, its use to define the model parameter set as part of the model development process is an extension of this approach. Our model is developed using a general-purpose programming language that enables a novel approach to calibration using a simulation optimisation approach. In doing so, computational overheads are reduced while promoting scalability and generalisability of the model.

## 3. The Hybrid Social Network and System Dynamics Model

In reality, individuals belong to a wide range of social networks. These can be face-to-face or facilitated by other communication modes, including social media. Moreover, two individuals can be contacts in a range of contexts (e.g., work, social, sport) and varying roles. This phenomenon (multiplexity) means that the network developed in the simulation represents an aggregate, combining elements from each of the networks that have an impact on obesity-related behaviour. This will be described in Section 3.1, followed by the description of the SD component in Section 3.2. The latter represents the internal decision processes of each individual within the network in relation to calorie consumption internalised within each agent.

### 3.1. The Social Network Model

Our social network model uses the emergent behaviour from an ABM. Each agent represents an individual situated in a broader social network having immediate neighbours within that network (network neighbourhood). This combination of stability and dynamicism is viewed as critical to the modelling of obesity-related social networks [14,34].

Our social network extends the work of Erbach-Schoenberg et al. [17] by adding agent heterogeneity and defined time parameters and further developing the social network constructs. Table 1 provides an overview of the sets, indices, and network parameters.

In the initialisation phase, agents are randomly allocated individual sets of characteristic variables from the data. An initial contact history is generated, and, in addition, each agent is randomly allocated a fixed position in a square two-dimensional ‘map’. The warm-up and simulation phases share an identical process (described below). The simulation phase includes processes for initialising new (adolescent) agents to add to the simulation and for removing ‘dead’ agents at the appropriate rates. Each iteration represents a month of model time. At each time step, each individual agent offers to link up with a number of other agents. Acceptance of these offers is contingent on:1.Propinquity—the tendency of individuals to form links with those whom they interact with frequently or are closest to.2.Homophily—the tendency of an individual to form links with those with whom they share some commonality.3.Memory—a stochastic probability defined by the number of times that an individual has previously linked with that individual.4.Each agent can only accept a fixed number of offers, defining the amount of ‘interaction time’ available for the agents.

Multiplexity is another feature of social networking; individuals belong to several networks, with potentially differing roles and expectations in each and with varying methods of interaction (face-to-face or electronic). However, within the context of disease modelling, there are little or no data with which to operationalize the concept. Our approach was to treat the model network as a proxy, representing the combined effect of these multiple networks and methods of interaction.

Figure 1 provides an example of the network at the end of a model run (10 years). Individuals with a healthy BMI of less than 25, overweight with a BMI greater than 25, strongly overweight with a BMI greater than 30, and morbidly obese individuals (with a BMI greater than 40) are represented by green, orange, red, and black nodes, respectively.

The process used to deliver the model is described in detail using pseudocode in Appendix B.

### 3.2. The System Dynamics Model

Figure 2 provides an overview of the behavioural model logic. The upper element uses a simplified version of the Theory of Planned Behaviour as first proposed in [24]. In the model, a decision to restrict calorie intake is triggered when the ‘Behaviour’ stock reaches a threshold; this in turn is driven by the ‘activation’ flow from the ‘Intention’ stock. Both the ‘Behaviour’ and ‘Intention’ stocks are subject to decay. The ‘activation’ flow is also potentially subject to a lag, the duration of which is one of the parameters explored in the optimisation process.

‘Intention’ is driven by three flows: ‘attitudes’, ‘norms’, and ‘pbc’ (perceived behavioural control):Attitudes—these are the sum of our knowledge, attitudes, and prejudices and result in the cognitive decision making processes described by [36], which in turn are moderated by the education level attained by the individual.Norms—are driven by input from the individual’s network neighbourhood, subject to it exceeding the satisficing value ($\theta_{sv}$). In our model, the individual uses the average BMI of the members of its network neighbourhood of the same gender, weighted according to the age of that individual.Perceived behavioural control (PBC)—is assigned a default value that is subsequently modified by the individual’s success or otherwise in losing weight.

The individual (agent) can take one of two states. State 1 represents the ‘normal’ situation in which consumption of calories is not limited. In State 2, calorie consumption is restricted (dieting). The transition from State 1 to State 2 is triggered once the threshold value for the ‘behaviour’ stock is reached and the agent perceives themselves as significantly overweight in relation to their immediate contacts in the network (network neighbourhood). The time spent in State 2 before transitioning back to State 1 is defined by a parameterised stochastic value and potentially further modified by individual success (or otherwise) in losing weight whilst in State 2.

The lower section of the model shown in Figure 2 shows weight change calculations. Weight change is influenced by ‘external factors’, such as advertising, cost of food, food availability and accessibility, social norms, representation in the media, etc. These are represented by a proxy, the national Average Per Capita Calorie Consumption (APCCC), as reported by the WHO, for each year of the simulation. Dietary reference values for the population taking into account age, weight, height, levels of activity, gender, and physiological state were obtained from the UK Scientific Advisory Council on Nutrition (SACN) [37]. The model uses these values to calculate an individual’s basal metabolic rate (BMR), and, in conjunction with data on physical activity levels (PAL), the person’s total energy expenditure (TEE). We have [38]:(1)TEE=PAL×BMR
(2)BMR=α+(β×height)+(γ×weight)
where α, β, and γ are coefficients that vary with age and gender.

Two additional elements are captured in the behavioural component of the model. Thomas et al. [39] point out that a conscious decision to reduce calorie intake (diet) is often only intermittently adhered to, resulting in the cyclical weight loss patterns and plateaus observed in practice. Hammond and Ornstein [40] also introduce the concept of ’Satisficing Behaviour’, describing the behaviour whereby an individual will only acknowledge a difference in BMI, and, hence, an imperative to act, if their BMI and the target BMI are separated by a value greater than some specified amount (satisficing value).

## 4. Hyper-Parameter Optimisation

Enns and Brandeau emphasise the importance of modelling the underlying contact structure (network structure) and dynamics when modelling the spread of a disease [41]. To achieve this, our model development phase used an optimisation algorithm to find the combination of network parameter values (that in turn define that structure) and population sub-groups to provide the most accurate forecast compared to real-world data. The possibility of large numbers of parameters, stochastic outputs, an undefined gradient function, and a computationally ‘expensive’ simulation presented a number of challenges. After some consideration, these were addressed using a simultaneous perturbation stochastic approximation algorithm (SPSA) [35] in conjunction with an appropriate loss function. SPSA is an iterative estimated gradient descent method, where, at each iteration, every parameter is modified simultaneously using a random perturbation vector, the gradient of the loss function is then estimated, and the parameters amended accordingly. The algorithm is recursive and takes the general form [42]:
(3)$${\hat{\theta }}_{k+1}={\hat{\theta }}_{k}-a_{k}{\hat{g}}_{k}\left({\hat{\theta }}_{k}\right)$$
where ${\hat{\theta }}_{k}$ is a vector of parameters, $a_{k}$ represents a scalar gain coefficient, ${\hat{g}}_{k}$ represents the gradient approximation, and *k* is the iteration count. This takes the general form:(4)$$g\left(\hat{\theta }\right)\equiv \frac{\delta L(\theta )}{\delta \theta }$$
where L(θ) represents a loss function. This uses the sum of squared errors and is calculated using the projected median and mean BMI for each gender and age group for each year and comparing them with the actual values for that time period.

The gradient function is calculated using the simultaneous random perturbation vector $\Delta_{k}$, and we use *c* as a scalar coefficient.
(5)$${\hat{g}}_{k}\left({\hat{\theta }}_{k}\right)=\frac{L\left({\hat{\theta }}_{k}+c_{k}\Delta_{k}\right)-L\left({\hat{\theta }}_{k}-c_{k}\Delta_{k}\right)}{2c_{k}}\left[\Delta_{k1}^{-1},\Delta_{k2}^{-1},\ldots \Delta_{kp}^{-1},\right]$$

The scalar coefficients are updated after each iteration:(6)$$a_{k}=\frac{a}{{\left(A+k+1\right)}^{\alpha }}$$
(7)$$c_{k}=\frac{c}{{\left(k+1\right)}^{\gamma }}$$

Following the recommendations in [35], we used the following parameters: a=0.16, A=100, c=0.1, α=0.602, and γ=0.101. Each implementation consists of 3000 iterations. A binomial distribution (1,−1) was used to realise the simultaneous perturbation vector $\Delta_{k}$.

### 4.1. Model Development

In total, three sets of data from the publicly available data published by HSE were used to develop the model. Two sets (2004–13, 2003–12) were used during the training process, an initial set to create the ‘base’ model and a development set to further refine it. The third data set (2002–11) was used for testing. These timeframes were chosen to maximise the data available for this process (the level of detail in the data published by HSE was subsequently reduced).

To reduce the impact of stochasticity, the initial population (for the training and development sets) used a fixed set of 1000 individuals (balanced for age, gender, and BMI) randomly selected from a data set of 4000. Thereafter, individuals were added randomly from a pool of randomly selected 16-year-olds from data for the correct year at an overall rate of 12 per year. ‘Deaths’ occurred at an overall rate of 7 per year (replicating the relevant birth and death rates from the period).

To further reduce the stochasticity of the model output, the final values for the parameters and loss scores in the training and development runs were determined by taking the median value of the last 100 iterations. The output from the initial descent run is provided in Figure 3.

After each descent run, the revised model was tested using the parameters generated by that run. For each iteration of the test, a random population was selected from a data set of 4000 agents for the start year (reflecting the relevant age and gender ratios within the population for that year). It was run for 10 years. This was repeated 1000 times, with a new population selected for each iteration. The results were then combined to produce a single set of data, which was compared with the real data for that 10-year period. The loss scores were also combined to provide a single representative value using median values.

In each run, we also examined the topographic structure of the social network.

### 4.2. Parameter Verification

The initial parameter set consisted of a set of single values representing each of the model parameters; in order to refine this, a ‘cascade’ strategy was implemented. After the initial training run (see Figure 3), each run involved examining the results from the previous run in conjunction with theory (on both social networks and TPB) in order to ascertain what parameter modifications might further improve the model. Modification was limited to splitting parameters by gender, age group, or both. These modifications were then implemented using the descent algorithm, and the resulting parameter values were then re-tested. Where the test results indicated a reduction in the loss function, the parameter changes were retained and implemented in subsequent descent runs.

The development process was also used to test the hypothesis that homophily (based on BMI) played a role in the formation of the network. The development process was carried out on two parallel tracks. In the first set of runs (‘Homophily’), BMI difference was allowed to affect the formation of the agents’ network neighbourhood; in the second (‘No Homophily’), it was not; otherwise, both used exactly the same parameter set. The results from each track were compared. Figure 4 provides the results, suggesting that incorporating homophily based on BMI leads to a more realistic model.

The parameters from this model were then further refined using the cascade process until the test score began to indicate over-fitting, at which point the parameter set was finalised. See Table 2.

The cascade process also highlighted the influence of agent gender. In the best-performing model, influence from an individual’s network neighbourhood was limited to those of the same gender as the individual.

### 4.3. Model Validation

To validate the model, the BMI figures (means and medians) generated by the final parameter set were compared with the actual figures for the same time period (2004–2013). The results are provided in Table 3. The negative values in the table indicate an issue where the model is forecasting values that are lower than the actual figures, whereas the positive values indicate that the forecasted values were higher.

The values show that there is some variation in the results for the highest and lowest age groups, where weight change models are generally less accurate. There is under-forecasting in the 21–30 age range for both genders, particularly for males.

## 5. Experimental Results

### 5.1. Scenarios

The experimental scenarios focused on investigating the impact of adolescent obesity and pressure from environmental factors (EF), the two factors deemed to have the most impact on individual changes in BMI. These were represented by the mean BMI of 16-year-olds joining the simulation as it runs and changes in the value of APCCC. To obtain realistic rates of change for both factors, the historical data were scanned to identify the fastest period of sustained growth, and this became the default rate of change in both the growth and decline scenarios for both factors. The different permutations for the initial scenarios are described in Table 4.

Furthermore, by comparing scenarios (2, 9) with the output from Scenario 1 under different conditions, we were able to gain qualitative insights into the relative impact of these factors. The striking feature of the results was how little impact variation the two factors had (typically less than 0.5%). Two additional scenarios were run, doubling the rates of change and comparing rising and falling adolescent BMI and rising and falling pressure from environmental factors. Apart from the obvious impact on the youngest age group of rapid growth/decline in adolescent BMI, the impact was still small, with minimal changes in mean BMI for all age groups and maximum changes in median BMI of less than 1%.

### 5.2. The Network Effect

A set of counterfactual comparisons were made to assess the level of impact of the network on individuals using Scenarios 1, 4, and 7. These were re-run with the network effect removed, and the results were compared with the original outputs. This created a complex picture: the Scenario 1 comparison suggested a broadly similar impact across the genders and age ranges, with none of the figures representing variations of more than 0.5%. The Scenario 4 comparison (falling adolescent BMI and pressure from environmental factors) provides a much clearer picture, with removal of the network effect reducing BMI across the board for the older female population and to a lesser extent the male population. However, the maximum gain in BMI is still less than 1% compared to the case with static adolescent BMI and levels of environmental influence. Conversely, for the two youngest age groups in each gender, there is an increasingly strong negative effect, with a maximum impact of 6%. This effect is almost precisely reversed with Scenario 9 (rising adolescent BMI and APCC), with figures of similar magnitude.

In summary, the network effect is most apparent in a dynamic rather than a static situation. For much of the population, the network effect seems to be relatively low (consistent with the inelasticity of the network already commented on) and acts to maintain the ‘status quo’ by mitigating changes in obesity levels, caused by rising or falling adolescent BMI and environmental factors. This effect seems to be reversed for the youngest age group, where the effect is to amplify the impact of BMI and environmental factors.

### 5.3. The Topography of an ‘Obesity’ Network

Table 5 provides the data for the proxy network generated in the best-fit model. The metrics describing the network generated by our model (clustering coefficient, transitivity, degree assortativity, and average degree) are broadly consistent with data collected from other social networks [11]. However, on further examination, the network model does exhibit some unusual features. BMI assortativity is positively correlated with age, while BMI and network neighbourhood size are inversely correlated. Although the mechanism is not clear, it seems that the magnitude of the network effect on an agent is inversely correlated with the level of BMI assortativity exhibited by that individual’s network neighbourhood.

#### External Factors

The values derived from the parameters also suggest that, as age increases, the impact of environmental factors such as food availability, advertising, and so on decreases for males. They also indicate that the impact of such environmental factors is higher on males than females. See Table 6.

### 5.4. Theory of Planned Behaviour

There were a number of parameters associated with the SD element of the model that delivered TPB.

Whilst there was a great deal of individual variation, the contributions of ‘norms’ and ‘attitude’ to the growth in the Intention stock were of the same order. The absolute contribution of ‘pbc’ was more difficult to interpret, but it was notable that the feedback loop that drives it had very little impact on the model.

The ‘norms’ flow is of particular interest since it partially defines the impact of the network neighbourhood on the individual. It is also affected by the satisficing number parameter; this acts as a threshold value (varying with the individual’s BMI) that, once exceeded, will trigger the norms flow. The final parameter value implied a satisficing number of 0 for an individual whose BMI is 20, rising to 1 for an individual whose BMI is 30 and 4 for an individual with BMI 40, implying that a person whose BMI is 40 would not recognise a difference in BMI until the difference was greater than 4 (36 or 44).

The values for $\theta_{norms}$ themselves were clearly higher for the youngest two female age groups compared to their male counterparts, suggesting greater susceptibility to the ‘network effect’. For older age groups, the picture was less consistent (Table 7).

### 5.5. Interpreting the Results

Our results suggest that homophily (based on BMI) plays a role in the spread of obesity through social networks. They also reveal that younger adults tend to have larger networks with little BMI-based assortativity, whereas older adults tend to have smaller networks with greater BMI assortativity. For the former, this creates local networks that are more amenable to the diffusion of information; for the latter, this is likely to be reduced. These findings support the speculation in the original research [4].

Our model performed best with gender-specific networks, that is, males being influenced by other males and females by other females. It also suggests that the influence of the network itself might be greater for some sub-groups than for others and that older males may also be less susceptible to external factors, such as food availability, advertising, and so on. The fact that obesity is correlated with age may also be relevant.

One can envisage a number of ways in which these effects could combine to form the ‘mechanism’ whereby social networks act to spread obesity. However, a detailed and credible hypothesis will require further research.

## 6. Discussion and Conclusions

In order to gain insight into the nature and level of impact of social networks on the spread of NCDs, we selected obesity as a study subject and built a hybrid simulation model incorporating appropriate social network and behavioural theory to model the effect. We used detailed public data to parameterise and validate the model. We ran a range of scenarios to develop the insights we have described.

Our key message is that the impact of social networks on NCDs varies by both age and gender, and that, in all likelihood, as the way we interact with our networks evolves, the impact too will evolve. Two obvious factors that will affect that impact are the increasing use of social media and the changes in face-to-face networking imposed by COVID-19 lockdowns.

### 6.1. Bias and Limitations

If the results were to be used in a practical forum, COVID-19 lockdowns and the continuing evolution in social media usage mean that the parameters of the network model used in our studies would have to be re-visited.

The behavioural model used here (TPB) may not be suitable for other NCDs, and the simulation may have to be modified with a different model used in the behavioural component.

To quote from George Box, *‘All models are wrong but some are useful’*. Our research has identified a set of parameter values that have not been experimentally verified; rather, they define a model that can achieve a certain level of accuracy in forecasting obesity and that is **likely** to represent many of the key relationships amongst the parameters.

The study has limitations. Translating social theory into mathematical relationships is always challenging, and, whilst the TPB has been widely applied in the literature and found to be a valid predictor of behaviour, there was very little information available on which to base the initial parameter settings, or with which to operationalise the model. The issue of variability in dieting behaviour is a complex one: it may be that the method chosen in the SD component to represent it was not sufficiently flexible to model the behaviours of the population as a whole. The data set available for the 16–20 age group was small in comparison to the remainder and suggested large variations in BMI. It is likely that this reflects the actual situation but could also be a consequence of the small size of the data set and collection issues. The PAL data used in the model are taken from (https://www.gov.uk/government/publications/sacn-dietary-reference-values-for-energy) (accessed on 1 March 2023). They involve taking one of three values dependent on BMI and age. This probably fails to adequately represent the variation in individual physical activity/exercise that is likely to occur within the network population. The simulation had issues with projections for the 21–30 age group. The 10-year projection for the final version was within 0.5 BMI (average error) for both genders in the remaining age groups.

More generally, the definition of an ‘obesity’ network is problematic. The multiple networks that an individual belongs to, their varying impact (conscious and unconscious) on the issue at hand, the diffuse boundaries of those networks, and the fact of constant evolution make it extremely challenging to define an obesity-specific network precisely. Instead, this research has identified a proxy network that approximates the impact of these in a single un-directed network. Thus, the network parameters derived from the model do not represent a single network but an amalgam of several different networks. This proxy network approximates the overall impact of the different elements of that amalgam during the timeframe used by the training/testing phases. The projections assume that this approximation remains valid for the period of the forecast. Over the lifetimes of the individuals represented in the simulation (and in the relevant literature), the process of building and maintaining such networks has changed radically. Much of the original research in this area used directed networks; the approach has necessitated the use of an un-directed network, making comparison of results complex.

### 6.2. A Generalisable Model

Generalisability can be considered in two domains: could this modelling approach be applied to other countries/regions to explore similar research objectives, and could this modelling approach be applied to explore the impact of social networks on other NCDs?

Regarding the first, there is no evidence to suggest that TPB is specific to a single culture or population segment, although it is likely that the relative impact of the parameter inputs may well vary with cultural differences. Given this, the optimisation process will ensure that the relevant parameters are ‘tuned’ appropriately. Similarly, for the network element, the topography may well vary according to location and culture, but, again, the optimisation process can address that. The critical issue with this level of generalisation is ensuring that there are enough data with which to train the model.

The second domain is more complex. The network component remains viable (given enough data to facilitate the optimisation process). The choice in paradigm for the behavioural/decision making component would have to be reviewed. For issues like smoking, drug use, and certain types of eating disorders, the model is unlikely to be the best option. In these circumstances, generalisability depends on the type of behavioural model best able to model the issue and its suitability for incorporation in an HS.

### 6.3. Managerial and Theoretical Insights

Historically, whilst there has been a great deal of concern about the rise in adolescent obesity levels  [43], in the United Kingdom, the National Health Service workload has primarily come from the older population. The data from the model suggest that, even if the current situation does not worsen, the workload from the younger male population is likely to increase significantly. The apparent greater susceptibility of the male population (particularly the younger age groups) to external factors may provide both an explanation of the effect and a lever to help mitigate the issue.

The model proved surprisingly inelastic in the presence of global interventions (falling adolescent BMI or a more supportive external environment) at what were designed to be the maximum realistic values for the population. Even when these rates were doubled, the impact was not great. This may at first seem to contradict the projections associated with initiatives such as a sugar tax [44,45], where significantly greater reductions in population BMI are expected. However, it is worth recalling that these initiatives anticipate much higher reductions in calorie intake and are focused on specific elements of the population. The levels used in the model are based on historical levels and thus could be considered realistic/achievable. This suggests that delivering whole-population initiatives may be challenging and it might be more productive to focus on specific issues and segments of the population.

One of the more striking effects was the relative impact of the network on the youngest age groups (24% of the total network population) and the more general influence of gender. When implementing weight loss strategies targeted at that age group, consideration could be afforded to leveraging these two effects to amplify the impact of the strategies. Valente [46] identifies four strategies in relation to network interventions in health care settings:Individual—identifying champions or opinion leaders.Segmentation—identifying cliques or groups on which to focus attention.Induction—deliberately creating interactions to spread information.Modification—adding new elements into the network or ‘re-wiring’ them to deliver the objectives.

The first three strategies increase the impact of a network on the individual. The last can be used either to reduce or increase the network impact and perhaps has more relevance to the older portion of the population. Given that an effect in the older populations was of a much smaller degree, initiatives focused on those age groups will have to consider whether the effort involved in manipulating the network is consistent with the expected return.

More fundamentally, this research suggests a variation in the network’s impact across the age groups that was not recognised in previous research. Previous research in this area used either the Framingham data [5] or the Longitudinal Study on Adolescent Health [47]. In both cases, the data either pre-date the existence of social media or fail to consider it in the collection methodology. The two data sets recorded information from two different age groups. Reference [47] recorded the data of adolescents between the ages of 14 and 19; Reference [5] recorded data for those over the age of 21. Unsurprisingly, their models use a single population model of network interaction. This research was able to address both issues and suggests that the nature of the network effect, the scale of impact, and the structure of the social network may differ significantly for different elements of that network. The key determinant seems to be age. A hypothesis consistent with this and the data used in the initial studies would be that the advent of social media has changed, and is perhaps continuing to change, the nature and impact of the ‘obesity’ network, and that this effect varies amongst the different sub-groups that make up the population.

### 6.4. Implementation and Future Research

An approach for implementing findings from health care simulations into policy is described by Araz et al. [48] in which the outputs of a simulation tool are used to develop a tabletop exercise that policymakers could use to design and refine effective policies. A similar approach would be an effective way to implement the findings from this research.

As previously discussed, multiplexity is a feature of social networking. However, the concept remains unacknowledged in any of the articles reviewed. Further research into this concept and alternative approaches would add value. Network realisations using emergent behaviour clearly have the potential to deliver sophisticated social network simulations but are relatively rare. Again, further research in this area would add value.

Focusing on examining the network effect as it applies to adolescents and young adults is an area where further research has the potential to add significant value. The findings suggest that this may have the potential for a significant impact and that the behaviour and level of impact may well have changed since the initial research was conducted.

Our findings also suggest that the impact of external factors varies across different age groups and genders. This has implications both for health behavioural models and more generally. Research to confirm the findings and quantify the level of impact would add value in the design of future health care interventions.

## Figures and Tables

**Figure 1 ijerph-20-06451-f001:**
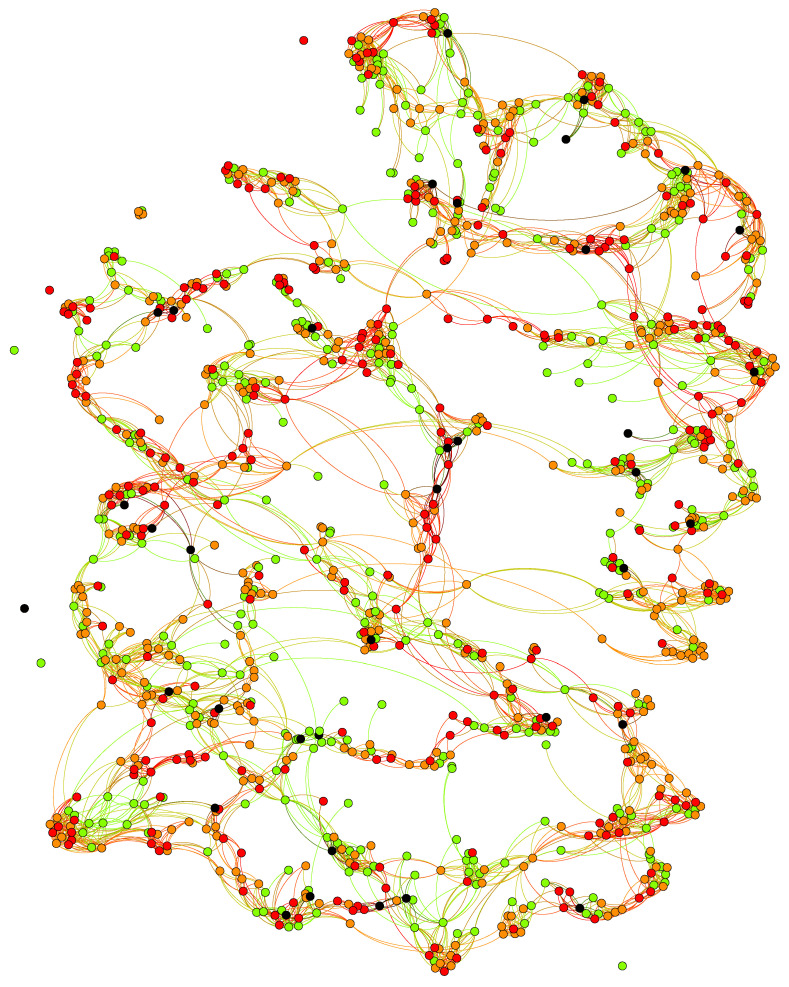
Social network model output.

**Figure 2 ijerph-20-06451-f002:**
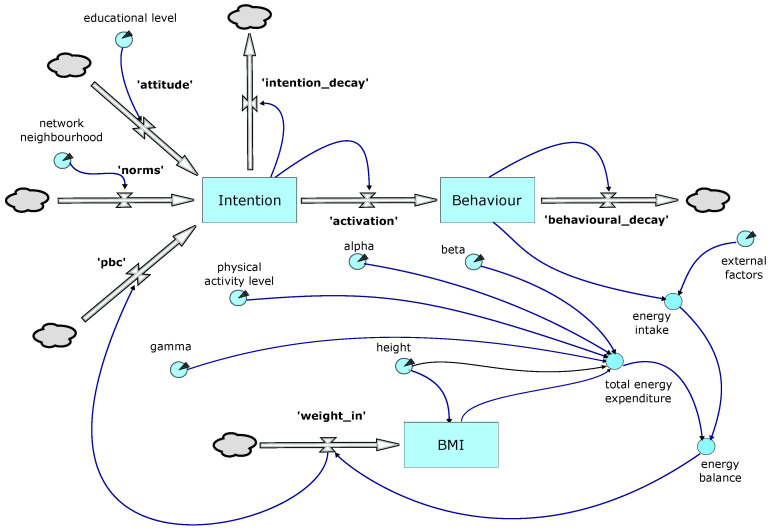
SD model.

**Figure 3 ijerph-20-06451-f003:**
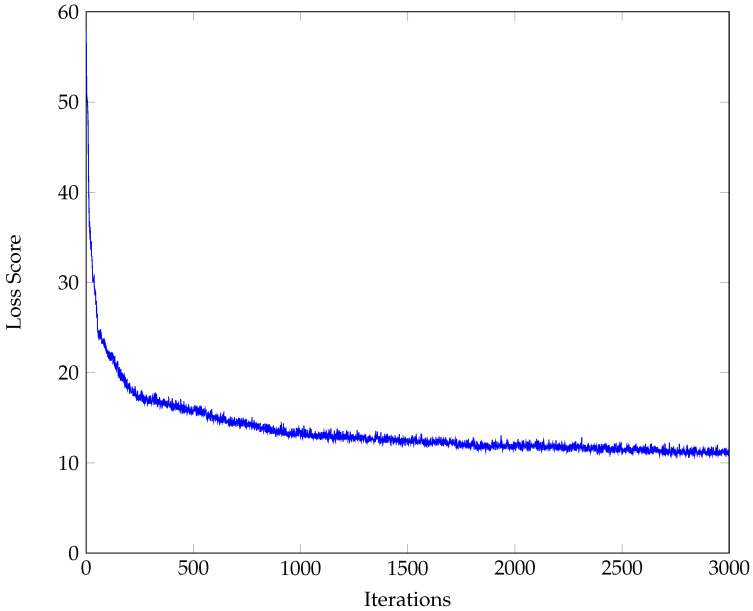
Loss score from initial descent run.

**Figure 4 ijerph-20-06451-f004:**
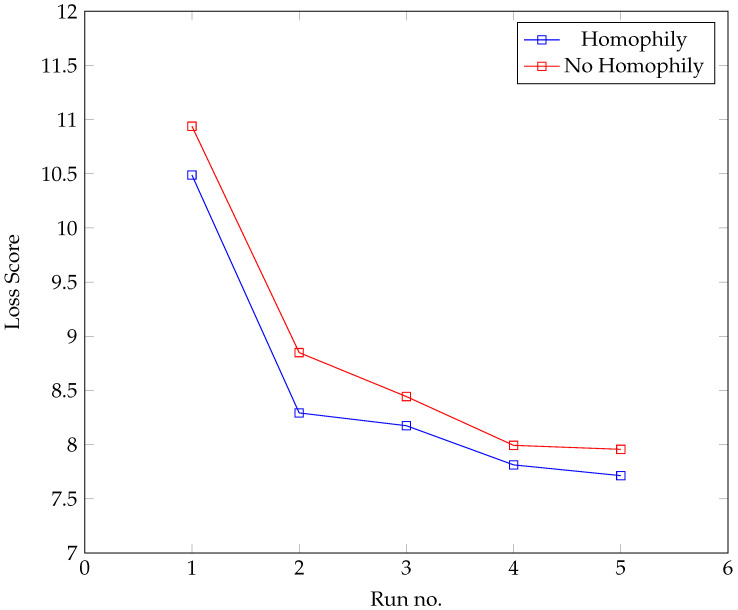
Test loss scores.

**Table 1 ijerph-20-06451-t001:** Social network parameters and sets.

Parameter	Description
Ra	Social range between agents (propinquity).
*S*	Maximum number of invitations accepted in each time slot (social resource).
*M*	A ‘memory’ network parameter in history of interactions.
*Z*	Network parameter describing the maximum value of p.x & p.y
*H*	Boolean variable (1,0) describing whether a relationship can or cannot exist between 2 agents (homophily).
*P*	The set of agents currently active in the network.
p.x & p.y	Characteristic variable: randomly allocated x and y axis positions for agent *p*.
p.bmi	Characteristic variable: current BMI value for agent *p*.
p.age	Characteristic variable: current age of agent *p*.
p.gender	Characteristic describing gender of agent *p*, 0,1.
$P_{p}^{a^{\prime }ted}$	The set of agent offers ($p^{\prime }\subset P$) accepted by an agent (*p*).
$P_{p}^{a^{\prime }tees}$	The set of agents ($p^{\prime }\subset P$) accepting an agent (*p*’s) offers.
$P_{p}^{lc}$	The list of agents ($p^{\prime }\subset P$) interacting with agent *p* in current time slot; may include repetitions.
$P_{p}^{nn}$	The set of agents ($p^{\prime }\subset P$) with which agent *p* is currently connected (network neighbourhood).
$P_{p}^{r}$	The set of agents ($p^{\prime }\subset P$) whose distance from *p* is less than Ra.
$P_{p}^{h}$	The set of agents ($p^{\prime }\subset P$) for which $H(p,p^{\prime })=1$.
R	Set of tuples $\left(p,P_{p}^{range}\right),P_{p}^{range}$ is the set of agents ($p^{\prime }\in P$) within social range of *p*.
H	Set of tuples $\left(p,P_{p}^{H}\right),P_{p}^{H}$ is the set of agents ($p^{\prime }\in P$) who share affinity with *p*.
$J_{p}^{prob}$	A set of tuples $\left(p_{\pi }^{\prime },p\right)\in J_{p}^{prob}$ with agent $p^{\prime }\in P$ and probability $\pi \in R_{\left[0,1\right]}$ describing the likelihood of accepting an offer from agent p∈P. $J_{p}^{prob}\left(p^{\prime }\right)$ returns the value π associated with $p^{\prime }$.
$J_{p}^{edge}$	A set of tuples $\left(p^{\prime },\alpha \right)\in J_{p}^{edge}$ with agent $p^{\prime }\in P$ and edge weight $\alpha \in R_{+}$ describing the edge weight between *p* and $p^{\prime }$ over the time period *M* (α= (no.of contacts in time H)/H). $J_{p}^{edge}\left(p^{\prime }\right)$ returns the value α associated with $p^{\prime }$
$J_{p}^{hist}$	A set of tuples $\left(\beta ,P_{p}^{lc}\right)$ with β∈Z, and $P_{p}^{lc}\in P$, describing the history of interactions of agent *p* with other agents $p^{\prime }\in P$. $J_{p}^{hist}\left(\beta \right)$ returns the set $P_{p}^{lc}$ associated with β.

**Table 2 ijerph-20-06451-t002:** Final parameter set.

Parameter	Component	Moderates	Split
$\theta_{norms}$	SD	‘norms’ flow	Split by age groups (6) and gender
$\theta_{pbc}$	SD	‘pbc’ flow	Single global value
$\theta_{attitude}$	SD	‘attitudes’ flow	Single global value
$\theta_{extFactor}$	SD	APCCC	Split by age groups (6) and gender
$\theta_{sv}$	SD	Satisficing value for ‘norms’ flow	Single global value
$\theta_{lag}$	SD	Lag between behaviour and Intention stocks	Single global value
$\theta_{dietTime}$	SD	Mean diet time for agents	Single global value
$\theta_{trigger}$	ABM	Threshold value for impact of BMI difference on homophily	Single global value
$\theta_{BMIadj}$	ABM	Impact of BMI on homophily	male → male, male → female,
			female → female, female → male.
$\theta_{mem}$	ABM	*M*	Single global value
$\theta_{range}$	ABM	Ra	Single global value

**Table 3 ijerph-20-06451-t003:** Model validation results; each data point represents the difference between actual BMI and forecast BMI.

	Male 16–20		Male 21–30		Male 31–45		Male 46–60		Male 61–75		Male 76+	
Year	Mean	Median	Mean	Median	Mean	Median	Mean	Median	Mean	Median	Mean	Median
1	−0.015	−0.022	−0.040	−0.046	0.001	0.016	−0.012	−0.008	−0.002	−0.009	0.022	0.019
2	−0.673	−1.554	−0.605	−0.866	0.594	0.963	0.214	0.291	0.428	0.209	1.156	0.780
3	−0.527	−0.635	−1.483	−1.659	0.164	0.477	−0.137	0.271	0.256	0.265	1.103	0.934
4	−1.189	−1.399	−1.474	−1.719	0.134	0.592	−0.096	0.162	0.015	0.050	1.047	0.931
5	−0.725	0.281	−1.005	−0.925	0.297	0.542	−0.173	0.067	0.304	0.501	1.093	0.941
6	−0.287	0.498	−1.463	−1.678	0.015	0.014	0.066	0.237	0.133	0.387	1.075	0.853
7	−0.779	−0.100	−1.387	−1.805	0.241	0.135	0.084	0.314	0.470	0.689	1.241	0.753
8	−0.445	0.722	−2.254	−2.263	−0.330	−0.426	0.101	0.400	0.283	0.643	0.907	0.580
9	−0.352	0.478	−1.929	−2.182	−0.310	−0.342	0.058	0.634	0.444	0.772	1.005	0.735
10	−1.482	0.040	−2.088	−2.329	−0.804	−0.692	0.280	0.738	0.646	0.967	1.403	1.281
	**Female 16–20**		**Female 21–30**		**Female 31–45**		**Female 46–60**		**Female 61–75**		**Female 76+**	
**Year**	**Mean**	**Median**	**Mean**	**Median**	**Mean**	**Median**	**Mean**	**Median**	**Mean**	**Median**	**Mean**	**Median**
1	−0.719	−0.743	−0.986	−0.809	0.060	−0.079	0.285	0.259	0.103	−0.267	−0.094	−0.421
2	−1.944	−1.377	−0.208	−0.183	−0.068	−0.328	0.331	0.419	−0.122	−0.585	0.707	0.463
3	−2.173	−1.155	−0.565	−0.562	0.030	0.084	0.377	0.409	0.255	−0.160	0.685	0.495
4	−1.473	−1.047	−0.316	−0.409	0.366	0.204	0.662	0.587	0.205	0.078	0.897	0.487
5	−0.580	−0.429	−0.759	−0.782	0.578	0.653	0.715	0.517	0.574	0.203	1.458	1.380
6	−0.475	−0.426	−0.445	−0.396	0.086	0.103	0.636	0.385	0.590	0.425	1.350	1.199
7	−0.904	−1.536	−1.125	−0.729	0.218	0.357	0.559	0.450	0.556	0.746	1.673	1.259
8	−0.068	−0.408	−1.327	−1.046	−0.123	−0.201	0.626	0.506	0.583	0.441	1.391	1.311
9	−0.803	−0.537	−0.967	−0.573	−0.216	−0.091	0.700	0.537	1.111	1.195	1.303	0.889
10	−0.408	−0.626	−1.525	−0.908	0.122	0.244	0.808	0.697	1.469	1.585	2.095	1.732

**Table 4 ijerph-20-06451-t004:** Scenario list: adolescent BMI versus environmental influence.

Scenarios:	1	2	3	4	5	6	7	8	9
Falling EF		*	*	*					
Static EF	*				*	*			
Rising EF							*	*	*
Falling Adolescent BMI				*		*			*
Static Adolescent BMI	*		*					*	
Rising Adolescent BMI		*			*		*		

The asterisks (*) for Static EF and Static Adolescent BMI in Scenario 1 mean that we incorporated both settings in this scenario.

**Table 5 ijerph-20-06451-t005:** Topography of an ‘Obesity’ network.

Metrics	Median	Mean	Std. Deviation	Definition
Clustering	0.67	0.67	0.01	The ratio of links (edges) in that node’s (agent’s) neighbourhood to the total possible links in that neighbourhood
Transitivity	0.63	0.63	0.01	The ratio between the observed number of closed triplets in the network and the maximum possible number of closed triplets.
Assortativity	0.34	0.34	0.05	The tendency of nodes to form connections between nodes of similar degree
Average Degree	9.35	9.35	0.19	Average number of connections for each node

**Table 6 ijerph-20-06451-t006:** BMI factor parameter values.

Age Range:	16–20	21–30	31–45	46–60	61–75	76+
Male	0.29	0.31	0.24	0.256	0.22	0.22
Female	0.20	0.22	0.22	0.20	0.21	0.20

**Table 7 ijerph-20-06451-t007:** Norm parameter values.

Age Range:	16–20	21–30	31–45	46–60	61–75	76+
Male	1.95	1.87	4.91	2.99	3.69	1.52
Female	2.84	3.85	1.02	4.99	0.87	3.725

## Data Availability

The obesity data used in this study can be accessed at: https://digital.nhs.uk/data-and-information/publications/statistical/health-survey-for-england/#past-publications, accessed on 16 July 2023. The dietary reference values used in this study can be accessed at: https://www.gov.uk/government/publications/sacn-dietary-reference-values-for-energy, accessed on 16 July 2023.

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
