# Peer review of "Understanding the Impact of Social Networks on the Spread of Obesity"

_ijerph, 2023, doi:10.3390/ijerph20156451_

Round 1

Reviewer 1 Report

This study constructed a new dynamic network model and conducted subgroup studies using HSE data sets. The study produced some very interesting results, which confirm the results of some recent studies.

This manuscript is rich in content. As an exploratory study, this paper reports many influencing factors based on the findings of the new model. However, the biggest problem of this paper is that it does not compare with the classical method, so it is suggested that the author increase the content of comparison with other network models. Please note that I am referring to the comparison of relevant network influence factors and network model performance, not to be descriptive in the discussion.

In the discussion, the researchers did not conduct a systematic in-depth discussion of the bias of this study, nor did they conduct a detailed discussion of the influencing factors and transmission factors of obesity based on the network findings. These contents should be added to ensure that the conclusion is reliable.

Author Response

Thank you for taking the time to review our paper.

Your comments are much appreciated and as a consequence we have:

  1. Included a more detailed explanation of the rationale for the network model choice (emergent ABM) as opposed to graphical or data-driven models.
  2. Restructured the manuscript too include a section on bias and limitations and included additional content specific to model bias.
  3. Included a section (‘Interpreting the results’) identifying the influencing and transmission factors

Thank you again for taking the time to review the paper we found your comments useful and positive.

Reviewer 2 Report

1.       The authors carried out research on a strategically important topic. Obesity is a distinctly topical disease that causes serious problems today and is expected to continue to cause problems in the future. The authors conducted their research using a particularly unique, innovative method.

2.       In terms of form and content, the manuscript meets the expectations of the journal editors.

3.       The authors could justify the social effects of the severity of obesity with a little more related and relevant literature within the Introduction chapter. This will give them a much better understanding of the real significance of their research.

4.       It would be worthwhile for the authors to create another flowchart that depicts the many methodologies used in the research and their connections. The authors have systematically incorporated many methods, their even more emphasized harmony should help the reader.

5.       The authors present the methods used in detail and analyze similar research from many other previous studies. This also greatly increases the value of their research.

6.       A particularly relevant part is the definition of limitations and further research directions. This also shows that the primary goal of the authors is not to create a complete theoretical model. The main goal is that the practical sphere can also use the results obtained by the authors and thereby contribute to the improvement of the obesity indicators of the population at the level of society as a whole.

7.       In my opinion, the authors have done particularly valuable work, the novelty of which is considerable. Exploiting the future research directions listed by the authors will be very important, for which I wish the researchers a very good job.

Author Response

Thank you for taking the time to review our paper.

Your comments are much appreciated and as a consequence we have:

  1. Added additional content to the Introduction describing the social significance of obesity on a global level.
  2. Added an appendix (referenced at the end of Section 2.) that describes the methodologies used and their place in the process.

Thank you again for taking the time to review the paper we found your comments useful and encouraging.

Reviewer 3 Report

The article is very interesting and quite scientifically based. The authors conducted a serious study. There are only a few minor comments. 1. In Table 1, the formula "Hm(p,p’)=1" is not sufficiently explained. 2. In Table 1, brackets (A, S) are used to describe tuples and (simultaneously) for the values of a two-element numeric set {0, 1}. Perhaps it would be better in the second case to use curly brackets {0, 1}? 3. Table 3 shows the differences between the actual and predicted BMI. However, in addition, the coefficient of determination should be calculated here.

Author Response

Thank you for taking the time to review our paper and your encouraging comments, we have implemented the notation changes you’ve suggested.

With regard to your comment about including a coefficient of determination whilst it is easy to calculate the values (0.74 and 0.80 for Mean and Median forecasts respectively).  Their validity is less clear, conventionally R squared values are used to evaluate and compare linear regression models, our model is not based on linear regression and we believe that therefore to include an R squared value (however favourable the number) would cause a great deal of critical comment.

Reviewer 4 Report

This paper investigate on the role of social networks, in theoretical terms, in chronic disease whose occurrence can depends on the individual dietary decision, as obesity. These decisions are based also on systems dynamics. Here, the authors study the age-groups and genders acting on Body Mass Index impacted by social factors.

The authors show findings demonstrating that yougest population receive a major impact than the older population.

The authors study the  the impact of social networks on the spread of obesity through a hybrid simulation (HS) in which the social network is replicated using an agent based model (ABM) and individual health behaviour is modelled using a system dynamics (SD) approach. Furhtermore they integrate the longitudinal data from Health Survey England (HSE).

Although this paper undoubtedly addresses a topic of great interest, even referring to very important literature studies such as the studies of Christakis and Fowler or Centola, it has many gaps: 

- The bibliographical references are too old. There are plenty of other papers and models that study social network effects and diffusion as a collective dynamic that impacts health issues. 

- Recently multiple studies have been done on the influence of social network in decisions involving Covid diffusion. 

- In addition, the authors make a very brief mention of the multiplex network, never defining it, which actually along with graph theory represents the best way to study diffusion and collective dynamics. 

- Then emphasizing the results of this work, which are related to groups and 'homophily among network nodes, they do not even mention community detection and clustering techniques. 

I suggest the following to the authors:

- Write in a clearer and more structured way the abstract

- Justify the choice of models used and explain in depth why multiplex networks and graph theory is not applied in this case. 

- Update the bibliographical references. 

Author Response

Thank you for taking the time to review our paper.

Your comments are much appreciated and as a consequence we have:

  1. Rewritten the abstract.
  2. Included a more detailed explanation of the rationale for the network model choice
  3. Expanded on our comments about multiplex networks. In terms of our approach to multiplexity, in our research we sought to construct a proxy network that represented the aggregated impacts of the range of contact networks and those facilitated by electronic media that combine to have an impact on NCD’s generally and obesity specifically. This approach was necessary because we were unable to access sufficient data for the population represented in our obesity data to identify and represent these separate networks credibly.
  4. Where it is feasible updated a number of the bibliographical references to more recent equivalents

Thank you again for taking the time to review the paper we found your comments useful and insightful.